# Analysis of Survival of Patients Hospitalized with COVID-19 in Espírito Santo, Brazil

**DOI:** 10.3390/ijerph19148709

**Published:** 2022-07-17

**Authors:** Juliana Rodrigues Tovar Garbin, Franciéle Marabotti Costa Leite, Luís Carlos Lopes-Júnior, Cristiano Soares da Silva Dell’Antonio, Larissa Soares Dell’Antonio, Ana Paula Brioschi dos Santos

**Affiliations:** 1Secretaria de Estado da Saúde do Espírito Santo, Special Epidemiological Surveillance Nucleus, Vitoria 29050-625, ES, Brazil; lissadellantonio@gmail.com (L.S.D.); anapaulabsantos86@gmail.com (A.P.B.d.S.); 2Graduate Program in Public Health, Federal University of Espírito Santo (UFES), Vitoria 29047-105, ES, Brazil; francielemarabotti@gmail.com; 3Hospital Sírio-Libanês, Instituto de Ensino e Pesquisa, São Paulo 01308-901, SP, Brazil; cristianosss@outlook.com

**Keywords:** hospitalization, COVID-19, SARS-CoV-2, survival, risk factors

## Abstract

Objective: To analyze the survival of patients hospitalized with COVID-19 and its associated factors. Methods: Retrospective study of survival analysis in individuals notified and hospitalized with COVID-19 in the state of Espírito Santo, Brazil. As data source, the reports of hospitalized patients in the period from 1 March 2020, to 31 July 2021 were used. The Cox regression analysis plus the proportional risk assessment (assumption) were used to compare hospitalization time until the occurrence of the event (death from COVID-19) associated with possible risk factors. Results: The sample comprised 9806 notifications of cases, with the occurrence of 1885 deaths from the disease (19.22%). The mean age of the group was 58 years (SD ± 18.3) and the mean hospital length of stay was 10.5 days (SD ± 11.8). The factors that presented a higher risk of death from COVID-19, associated with a lower survival rate, were non-work-related infection (HR = 4.33; *p* < 0.001), age group 60–79 years (HR: 1.62; *p* < 0.001) and 80 years or older (HR = 2.56; *p* < 0.001), presence of chronic cardiovascular disease (HR = 1.18; *p* = 0.028), chronic kidney disease (HR = 1.5; *p* = 0.004), smoking (HR = 1.41; *p* < 0.001), obesity (HR = 2.28; *p* < 0.001), neoplasms (HR = 1.81; *p* < 0.001) and chronic neurological disease (HR = 1.68; *p* < 0.001). Conclusion: It was concluded that non-work-related infection, age group above or equal to 60 years, presence of chronic cardiovascular disease, chronic kidney disease, chronic neurological disease, smoking, obesity and neoplasms were associated with a higher risk of death, and, therefore, a lower survival in Brazilian patients hospitalized with COVID-19. The identification of priority groups is crucial for Health Surveillance and can guide prevention, control, monitoring, and intervention strategies against the new coronavirus.

## 1. Introduction

COVID-19 (Coronavirus Disease 2019) has gained prominence as it has become the most widespread acute respiratory infection with global distribution and is characterized as a disease caused by the novel coronavirus SARS-CoV-2. The incubation period varies from 1 to 14 days, with a median of 5 to 6 days [1] and is responsible for variable symptomatology ranging from mild signs to severe conditions requiring hospitalization [2]. The most common symptoms in hospitalized patients are fever, dyspnea, and ground-glass opacity demonstrated by imaging tests [3]. 

The risk for a more severe form of the disease was associated with comorbidities such as diabetes mellitus, hypertension and smoking [4]. Similarly, sociodemographic aspects such as advanced age (80 years or older), male sex, race/non-white color and being a public hospital user have been shown to contribute to a higher risk of death [5,6,7]. In addition, it should be noted that a more critical course of the disease may promote increased hospital admissions and hospital length of stay [8], generating demands for health systems. 

In this sense, by identifying characteristics that favor a more severe outcome, the studies of survival analysis have been useful since they report the time elapsed between the onset and the event of interest [9]. After electing hospital admission and/or the onset of the symptoms of COVID-19 as the initial event and death as the final event, researchers pointed out that patients with a worse survival from COVID-19 were those with dyspnea, pneumonia, ground-glass opacity, admitted to the Intensive Care Unit (ICU) with the support of mechanical ventilation, with chronic kidney disease, in addition to being of advanced age and belonging to the male sex [7,10,11,12]. 

In Brazil, from 26 February 2020, to 4 December 2021, a total of 22,138,247 cases and 615,570 deaths from COVID-19 were confirmed, with an incidence rate of 10,454.6 cases per 100,000 population and mortality of 290.7 deaths per 100,000 population [13]. 

When monitoring cases of Severe Acute Respiratory Syndrome (SARS) in the country, the Influenza Epidemiological Surveillance System (SIVEP-Gripe) recorded 2,812,789 hospitalizations in the same period. During the year 2021, from the total of 1,635,448 hospitalized SARS cases, 71.9% (1,176,355) were COVID-19 infections, and the most affected region was the Southeast, with 48.5% of the confirmed cases (571,034) [13]. 

The state of Espírito Santo, also located in the aforementioned region, reported 622,498 confirmed cases for COVID-19 and 13,208 deaths [14] in the same period. In the year 2021 alone, there were 7126 cases of SARS requiring hospitalization [15]. However, the understanding of these data through survival studies has not yet been elucidated. Thus, the evidence resulting from survival studies is important to establish the epidemiological and clinical profile of COVID-19 in the hospital setting. Therefore, the objective of the present study is to analyze the survival of patients hospitalized due to COVID-19 and its associated factors. 

## 2. Materials and Methods

This is a retrospective study of survival analysis in notified and hospitalized individuals with COVID-19 in the state of Espírito Santo, the smallest and least populous state in the southeastern region of Brazil, with a population of 4,108,508 inhabitants distributed in 78 municipalities [15]. According to a ranking by the nongovernmental organization Open Knowledge Brazil (OKBR), it was considered the most transparent state in the disclosure of data on COVID-19 in Brazil, [16]. 

The data used were from the State Secretariat of Public Health of Espírito Santo (SESA-ES) through the e-SUS Health Surveillance information system (e-SUS-VS) [17]. The notifications of inpatients in the period from 1 March 2020 to 31 July 2021, were used as data source. The adopted inclusion criterion was confirmation of the diagnosis for COVID-19 and hospitalization in Espírito Santo resulting from this diagnosis. Exclusion criteria were hospitalizations before the onset of symptoms, absence of the date of discharge/outcome, and hospitalization interval shorter than 24 h. 

For survival time, the start date was considered the date of admission, and the end date was the date of death (for failures), the date of discharge, and the date of death from other causes (censoring data). 

Sociodemographic variables were used, including information to know whether the patients were health professionals and whether they had a work-related infection. The state of Espírito Santo is the only one in Brazil that has this last variable in its notification form. Information on clinical variables and comorbidities was also used. The programs used in the analyses were IBM SPSS Statistics version 24 and STATA version 14. Multiple logistic regression with the forward variable selection method associated the outcome (death from COVID-19) with possible influencing factors. We only included in the model confounding variables that could be independently associated with death. Therefore, signs and symptoms were not included. The factors that remained in the final model were used for further analyses. The permanence in the model was given by a *p* value of < 0.05. The Kaplan-Meier estimator was used to observe the failure event (death) in addition to estimating the mean of hospital length of stay of patients in each factor. The Log-rank test was used to compare the equality of the survival curves. 

Cox regression in conjunction with proportional hazards assessment (assumption) was used to compare the hospital length of stay until the occurrence of the event (death from COVID-19) associated with possible risk factors. When the proportionality assumption was not met, the extended Cox model, commonly called the model with time-dependent covariates, was used. The alpha level of significance used in all analyses was 5%. 

Ethical approval was obtained by the Research Ethics Committee under number 5.180.941 on 20 December 2021.

## 3. Results

The sample totaled 9806 notifications of confirmed cases of hospitalized COVID-19 patients in the period from 1 March 2020 to 31 July 2021, with the occurrence of 1885 deaths from the disease (19.22%). The mean age of the group was 58 years (SD ± 18.3) and the mean of the hospital length of stay was 10.5 days (SD ± 11.8). 

Characterization of the inpatients was based on the following information: individuals aged 0–59 years (52.4%) were the most prevalent age group, 4 out of 10 patients had not finished elementary school, were not pregnant, and belonged to a non-white race/color. Regarding sex, about 54% were men. Regarding the place of origin of the notification of the case, the tertiary care level was responsible for 62.5% of the sample, and almost all the notifications were from people without disabilities (PWD) (96.5%) and non-homeless people (97.8%). As for data on residence and hospitalization, the Metropolitan Health Region (Regional de Saúde Metropolitana), a division of the national health system, accounted for the majority of the notifications of the cases, with 64.4% and 63.9%, respectively, with 86.7% of the inpatients living in the urban area. Regarding work, 89.3% were not health professionals and 75.8% had no work-related infection. Regarding clinical aspects, they did not meet the criteria for ICU admission (50.2%) and the most observed confirmation criterion was the one provided by confirmatory laboratory tests (97%). Regarding symptoms, cough (66.5%) and fever (58.5%) were predominant, and, as comorbidity, diabetes mellitus stood out (20.5%) (Table 1). 

In the crude and adjusted multiple logistic regression after the odds ratios were calculated, the clinical and sociodemographic factors associated with death from COVID-19 were the following: absence of a work-related infection (OR = 3.8; 95%CI = 1.89–7.65) when compared to its presence; age groups 60–79 (OR = 3.59; 95%CI = 2.94–4.39) and 80 years or older (OR = 8.3; 95%CI = 6.40–10.77) when compared to age group up to 59 years; education in the categories illiterate (OR = 1.42; 95%CI = 1.06–1.89), high school (OR = 1.68; 95%CI = 1.21–2.32) when compared to higher education, and of comorbidities such as chronic cardiovascular disease (OR = 1.73; 95%CI = 1.45–2.07), chronic kidney disease (OR = 2.66; 95%CI = 1.68–4.19), immunodeficiency (OR = 4.54; 95%CI = 1.78–11.55), smoking (OR = 2.73; 95%CI = 1.90–3.91), obesity (OR = 8.33; 95%CI = 6.68–10.38), neoplasms (OR = 5.40; 95%CI = 3.27–8.93) and chronic neurological disease (OR = 5.83; 95%CI = 4.0–8.5) when compared to their absence (Table 2). 

Based on this rationale, these were the variables chosen for the survival analysis. The proportionality assumption for the Cox model was tested using the Pearson’s chi-square test (Table 3). 

Table 4 presents the results of the univariable survival analysis using the Kaplan-Meier method. The result of the log-rank test for comparison of the categories of the study variables showed that the probability of survival of individuals hospitalized with the disease with a work-related infection at the end of the observation period was 94.3%, while the result was 78.4% for individuals who did not have the infection. 

When considering the age group, a lower probability of cumulative survival is observed with advancing age groups (up to 59 years—91.2%; 60–79 years—73.7%; and 80 years or older—57%). As for schooling, people who had a high school education had an 82% probability of cumulative survival, while illiterate people had 72.5%. 

Regarding comorbidities, in individuals without comorbidities, the cumulative survival was 88.3% for chronic cardiovascular disease, 81.3% for chronic kidney disease, and 81.7% for smoking, 83.7% for obesity, 81.4% for neoplasms and 82.3% for chronic neurological disease, while in individuals with comorbidities, the cumulative survival was 69.9%, 59.2%, 53.4%, 53.3%, 47.4%, and 46.3%, respectively. For the overall variables mentioned, there were statistically significant differences in the survival curves between the groups (*p* < 0.05). 

In the multivariable analysis (Table 5), after adjustment, the factors with a higher risk of occurrence of deaths from COVID-19 (associated with a lower survival) were non-work-related infection (HR = 4.33; *p* < 0.001), age groups 60–79 (HR = 1.62; *p* < 0.001) and 80 years or older (HR = 2.56; *p* < 0.001), presence of chronic cardiovascular disease (HR = 1.18; *p* = 0.028), chronic kidney disease (HR = 1.5; *p* = 0.004), smoking (HR = 1.41; *p* < 0.001), obesity (HR = 2.28; *p*< 0.001), neoplasms (HR = 1.81; *p* < 0.001 and chronic neurological disease (HR = 1.68; *p* < 0.001)

Figure 1 and Figure 2 show the difference in survival between the variables with statistical significance in the Cox Regression Model. It can be seen that the group with no work-related infection had a shorter survival time. The same occurred with the group belonging to the age group 80 or older and chronic cardiovascular disease, chronic kidney disease, smoking, obesity, neoplasms and chronic neurological disease (*p* < 0.05). 

## 4. Discussion

The present study aimed at analyzing the survival of patients hospitalized with COVID-19 and identifying its associated factors. In general, we found that the group of patients who did not have a work-related infection had a shorter survival time. Similarly, the group of patients belonging to the age group 80 and older with presence of chronic cardiovascular disease, chronic kidney disease, chronic neurological disease, smoking, obesity and neoplasms were associated with a shorter survival time. 

To the best of our knowledge, after extensive review of the scientific literature to date, it was noted that this is the first study conducted with the objective of analyzing the survival of patients hospitalized due to COVID-19 in the state of Espírito Santo, Brazil having as data source the state’s official Health Surveillance Information System, the e-SUS-VS. 

We found that 1885 patients hospitalized due to COVID-19 progressed to death from the disease, which corresponds to about 19% of the total notifications of patients confirmed and hospitalized for COVID-19 (N = 9806) in the period from March 2020 to July 2021. This finding is similar to that found in a historical cohort from Brazil, in which approximately 17% of patients hospitalized due to COVID-19 progressed to death [18]. Regarding the average of hospital length of stay, the results show a mean of 10.5 days (SD ± 11.8). A survey conducted in São Paulo, Brazil, showed an average of hospital length of stay of nine days. Considering only critically ill patients, the average of hospital length of stay in the ICU was 15 days and the average total of hospital length of stay was 22 days, while for patients who did not require ICU, the average of hospital length of stay was seven days [19]. Another similar study reported a case of hospital length of stay of 10 days in men and 15 days in women with mechanical ventilation time similar to the hospital length of stay in both sexes [20]. 

As regards the risk factors for death from COVID-19, the following factors were associated with shorter survival: age group 60–79 years and 80 years or older, non-work-related infection, patients with comorbidities, especially the cardiovascular ones, chronic kidney disease, chronic neurological disease, smoking, obesity and neoplasms. 

It is important to consider that the pandemic caused by COVID-19 that intensely ravaged the world impacted individuals in their different life cycles. In terms of mortality, the elderly population was the most affected. In several countries, it was found that persons aged 60 years or older are more vulnerable to the disease [3,12,20]. 

Individuals aged 60–79 years presented a relative risk of lower survival of 1.62 (*p* < 0.0001) compared to individuals aged 0–59 years, while a risk of 2.56 (*p* < 0.001) was observed in the age group 80 years or older. This finding corroborates other studies [3,11,12,18]. Elderly patients with COVID-19 are subject to progress to a severe disease, as they are more susceptible to a multisystem organ dysfunction and even organ failure. Furthermore, it was observed that the presence of lung lesions on imaging examinations and the number of white blood cells and neutrophils in laboratory examinations were significantly higher in the older patients group compared to the young and middle-aged group, thus suggesting that the former is also more prone to acquire a bacterial coinfection [11,21,22,23]. 

A large number of patients have cardiovascular disease as a comorbidity or develop cardiovascular dysfunction during the pathological process after SARS-CoV-2 infection [24]. The findings of the current study showed that chronic cardiovascular disease, including hypertension, is a factor associated with a higher risk for death and a shorter survival among hospitalized patients, thus corroborating the literature [25]. The impact of COVID-19 on the cardiovascular system is worth noting, as individuals with pre-existing disease are among the most affected by the disease and have a more severe clinical course [24,25,26,27,28,29,30]. 

To understand the dynamics of transmission and analyze the presence of a possible relationship between illness and work, the state of Espírito Santo, Brazil, created a variable called “Work-Related Infection” for e-SUS-VS. The results pointed out that although most of the sample (75.8%) had not shown the relationship between illness and work, when associating it with the outcome death from COVID-19, it was observed that non-work-related infections had a relative risk of 4.33 (*p* < 0.0001) of a worse survival when compared to professionals who had a work-related infection. Since the state of Espírito Santo is the only one that has this field to fill out, it is a challenge to compare it to other states in this regard. 

In spite of the fact that the Occupational Safety and Health Administration of the United States of America stated that professional categories other than health care have a lower risk of becoming ill [31], a document published in the state of Bahia, Brazil [32], raised the problem of the official notification systems which are not very sensitive to identifying work-related deaths, and estimated a high number of underreporting [32,33,34]. The work-related infections are based on the assumption that the determinants of this illness are related to the physical and social environment of work. Thus, a variety of diseases can be related to work, as long as they find favorable conditions for their occurrence in the environments. However, there is a major limitation in official notification systems, which are insensitive to identifying work-related infections and deaths. Thus, it is important to highlight the need of a complementary epidemiological investigation to identify situations of exposure to SARS-Cov-2 at work, which was not possible to accomplish in the current study. Indeed, we consider this to be a limitation of our study. 

Our findings pointed out that obesity, when present, is 2.3 times more likely to have a worse survival among people affected by COVID-19. The correlation between obesity as a risk factor for COVID-19 was not initially described in some countries such as China, Italy and the United States [35], despite the fact that obesity is closely associated with some diseases that indicate a high morbidity and mortality from this new infection, such as heart disease, diabetes and respiratory conditions [36]. Studies began to emerge linking obesity as a risk factor for worse outcomes of COVID-19 [37,38]. Furthermore, a study [38] has reported a risk of death due to COVID-19 up to four times higher in people with obesity, in addition to those 85% of patients with a BMI ≥ 40 kg/m^2^ who require invasive mechanical ventilation. Obesity is associated with a chronic state of meta-inflammation with systemic implications for immunity in which antiviral responses are delayed and insensitive, with a weakened immune response that may trigger the worsening of the disease [39].

It was also evidenced that the habit of smoking increases by 1.4 times the chance of a worse survival when affected by SARS-CoV-2. It is known that smoking predisposes to the development of several diseases that increase the severity of COVID-19 and therefore can be considered a risk factor [40]. Studies indicate that besides increasing the chances of hospitalization, smoking worsens the clinical picture, influences the progression of pneumonia and worsens the prognosis of the disease [41,42], in addition to increasing the risk of death from SARS-CoV-2 infection [43]. 

Smokers are 2.4 times more likely to be admitted to the Intensive Care Unit, need invasive mechanical ventilation or die compared to non-smokers [44]. It is known that tobacco can cause chronic obstructive pulmonary disease that promotes increased expression of the Angiotensin-2 Converting Enzyme that facilitates the entry of SARS-CoV-2 into cells. Thus, the lung damage caused by tobacco use increases the risk of COVID-19 and progression to severe disease [45].

The concern with patients with chronic kidney disease (CKD) and hospitalized due to SARS-CoV-2 infection has also been described in the literature, as this association leads to increased hospital mortality and a worse clinical outcome [46], which corroborates our findings in which a risk of 1.50 (*p* < 0.001; 95%CI = 1.14–1.99) was demonstrated. Similarly, a study conducted with a sample of 3391 patients with COVID-19 in New York showed that patients with chronic kidney disease had a higher risk of mortality (RR = 2.51 [95%CI: 1.82–3.47], *p* < 0.001) and intubation (RR = 2.05 [95%CI: 1.40–3.01], *p* < 0.001) [47]. Therefore, it is suggested that a history of CKD should be taken into account during risk stratification of patients with COVID-19 [46].

Patients with neoplasms are more susceptible to infections in general, due to the systemic immunosuppressive state caused by the treatments and are likely to have an increased risk of a worse prognosis. In China, patients with various types of cancer, particularly hematologic and lung cancers, were more likely to develop serious complications from COVID-19 [48] with the advanced stage of the disease being an even greater aggravating factor [49].

It is also known that people with pre-existing neurological diseases were also unequally affected by COVID-19, as they suffered from the exacerbation of their underlying disease. Previous studies have shown that chronic neurological disorders are independently associated with increased mortality in hospitalized COVID-19 patients (HR = 2.13; 95%CI: 1.38–3.28) [50,51].

Some limitations should be considered. Although it is the most common data source for studies with survival analysis, a secondary bank is subject to information bias. It is believed that, due to the high turnover during the pandemic, untrained professionals may have failed to understand the variables on the notification form when filling it out, and thus promoted inconsistencies in the information, especially those related to work-related infection. Another relevant limiting factor was the incompleteness of the variables, especially those related to the patient’s hospitalization, such as hospital bed and dates of hospitalization and discharge, which were responsible for the exclusion of most of the sample. However, even with the limitations pointed out, most of the findings are consistent with the world literature and can contribute to epidemiological and survival analyses of COVID-19, extrapolating the state of Espírito Santo, Brazil. We suggest the conduction of other studies on the subject that can direct efforts to the understanding of a possible relationship between illness and work. Moreover, the in-service training of professionals, in a systematic and continuous way, is essential for reducing failures in filling out the notification and improving data collection. 

As a strength and implications of this study, our results provide further evidence that there is an increased risk of death for elderly patients with chronic cardiovascular disease, chronic kidney disease, smoking, obesity, neoplasms and chronic neurological disease, as it results in a worsening of the patient’s condition. Undoubtedly, the identification of factors associated with the severity of the disease can effectively facilitate the management and provide assistance in order to improve the patients’ survival.

## 5. Conclusions

This study represents the first survival analysis on COVID-19 in the state of Espírito Santo, Brazil. Through it, it was possible to trace the factors associated with death and better understand the disease, especially the death of hospitalized patients. In summary, given the heterogeneity of the population, special attention shall be given to the elderly population, the patients with comorbidities, especially the cardiovascular ones, chronic kidney disease, chronic neurological disease, smoking, obesity and neoplasms. Thus, the identification of priority groups can direct the implementation of more specific prevention and control strategies, such as those that protect the elderly, people with comorbidities and prioritize testing for early detection of positive cases. Similarly, it is also important to disseminate the information on the most susceptible groups to the media, care networks, community, and other sectors beyond the health area.

## Figures and Tables

**Figure 1 ijerph-19-08709-f001:**
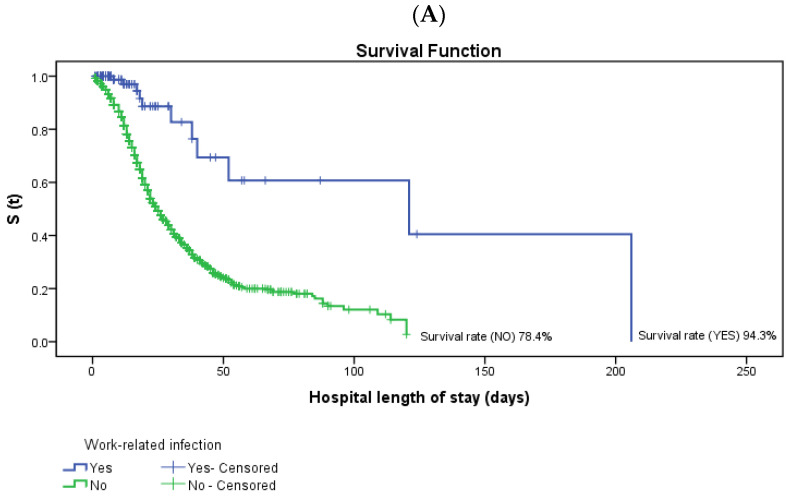
Survival curves of patients hospitalized with COVID-19 in Espírito Santo, Brazil. Log-rank test was used for comparison of equality of survival curves of sociodemographic variables in the period from 1 March 2020 to 31 July 2021. (**A**) Survival function according to the hospital length of stay of patients hospitalized with COVID-19 in Espírito Santo, Brazil, according to the independent variable work-related infection. (**B**) Survival function according to the hospital length of stay of patients hospitalized with COVID-19 in Espírito Santo, Brazil, according to the independent variable age group.

**Figure 2 ijerph-19-08709-f002:**
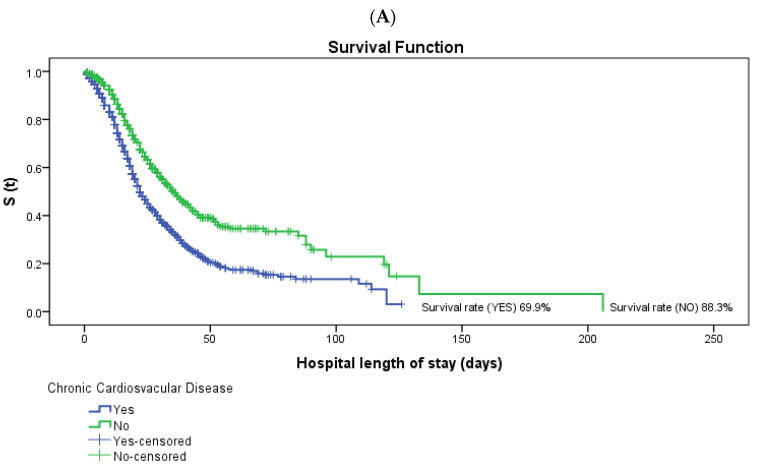
Survival curves of patients hospitalized with COVID-19 in Espírito Santo, Brazil. Log-rank test was used for comparison of equality of survival curves of clinical variables in the period from 1 March 2020 to 31 July 2021. (**A**) Survival function according to the hospital length of stay of patients hospitalized with COVID-19 in Espírito Santo, Brazil, according to the independent variable chronic cardiovascular disease. (**B**) Survival function according to the hospital length of stay of patients hospitalized with COVID-19 in Espírito Santo, Brazil, according to the independent variable chronic kidney disease. (**C**) Survival function according to the hospital length of stay of patients hospitalized with COVID-19 in Espírito Santo, Brazil, according to the independent variable smoking. (**D**) Survival function according to the hospital length of stay of patients hospitalized with COVID-19 in Espírito Santo, Brazil, according to the independent variable obesity. (**E**) Survival function according to the hospital length of stay of patients hospitalized with COVID-19 in Espírito Santo, Brazil, according to the independent variable neoplasms. (**F**) Survival function according to the hospital length of stay of patients hospitalized with COVID-19 in Espírito Santo, Brazil, according to the independent variable chronic neurological disease.

**Table 1 ijerph-19-08709-t001:** Sociodemographic and clinical characterization of hospitalized patients with COVID-19 in Espírito Santo, Brazil, in the period from 1 March 2020 to 31 July 2021.

		Death due to COVID-19				Total	
		No		Yes			
		N	%	N	%	N	%
**Sociodemographic variables**	**Categories**						
Age range	0–59 years	4681	59.1	453	24	5134	52.4
Education	Elementary school	1599	38.9	594	48.9	2193	41.2
Pregnant women	No	3550	44.8	847	44.9	4397	44.8
Race/Color	Nonwhite	3577	45.2	894	47.4	4471	45.6
Sex	Male	4278	54	1034	54.9	5312	54.2
Place of Notification	Tertiary	4861	61.4	1265	67.1	6126	62.5
PCD	No	7628	96.3	1838	97.5	9466	96.5
Homeless	No	7715	97.4	1876	99.5	9591	97.8
Household Health Region	Metropolitan	4792	60.5	1527	81	6319	64.4
Household Health Region	Metropolitan	4726	59.7	1537	81.5	6263	63.9
Zone	Urban	6778	85.6	1727	91.6	8505	86.7
Health Professional	No	7013	88.5	1742	92.4	8755	89.3
Work-related Infection	No	5822	73.5	1609	85.4	7431	75.8
**Clinical Variables**							
Fever	Yes	4542	57.3	1197	63.5	5739	58.5
Difficulty Breathing	Yes	2824	35.7	1188	63	4012	40.9
Flapping Wing Nose	Yes	69	0.9	36	1.9	105	1.1
Intercostal Tachycardia	Yes	141	1.8	46	2.4	187	1.9
Cyanosis	Yes	51	0.6	41	2.2	92	0.9
O_2_ Saturation < 95%	Yes	1745	22	1020	54.1	2765	28.2
Coma	Yes	20	0.3	22	1.2	42	0.4
Coughing	Yes	5208	65.7	1287	68.3	6495	66.2
Sputum Production	Yes	437	5.5	108	5.7	545	5.6
Nasal or Conjunctival Congestion	Yes	946	11.9	201	10.7	1147	11.7
Runny Nose	Yes	2034	25.7	429	22.8	2463	25.1
Sore Throat	Yes	1474	18.6	297	15.8	1771	18.1
Difficulty Swallowing	Yes	254	3.2	62	3.3	316	3.2
Diarrhea	Yes	1171	14.8	252	13.4	1423	14.5
Nausea/Vomiting	Yes	1057	13.3	218	11.6	1275	13
Cephalea	Yes	3079	38.9	555	29.4	3634	37.1
Irritability/Confusion	Yes	106	1.3	37	2	143	1.5
Adynamia/Weakness	Yes	2403	30.3	805	42.7	3208	32.7
Pharyngeal Exudate	Yes	64	0.8	17	0.9	81	0.8
Conjunctivitis	Yes	40	0.5	7	0.4	47	0.5
Convulsion	Yes	22	0.3	9	0.5	31	0.3
Loss of Sense of Smell	Yes	712	9	148	7.9	860	8.8
Loss of Taste	Yes	804	10.2	156	8.3	960	9.8
**Comorbidities**							
Chronic Pulmonary Disease	Yes	336	4.2	197	10.5	533	5.4
Chronic Cardiovascular Disease *	Yes	2816	35.6	1208	64.1	4024	41
Chronic Kidney Disease	Yes	140	1.8	100	5.3	240	2.4
Chronic Liver Disease	Yes	20	0.3	31	1.6	51	0.5
Diabetes Mellitus	Yes	1368	17.3	643	34.1	2011	20.5
Immunodeficiency	Yes	41	0.5	27	1.4	68	0.7
HIV Infection	Yes	30	0.4	9	0.5	39	0.4
Neoplasm (Solid or Hematological Tumor)	Yes	58	0.7	40	2.1	98	1
Smoking	Yes	177	2.2	155	8.2	332	3.4
Bariatric Surgery	Yes	17	0.2	1	0.1	18	0.2
Obesity	Yes	505	6.4	442	23.4	947	9.7
Tuberculosis	Yes	4	0.1	7	0.4	11	0.1
Neoplasms	Yes	92	1.2	101	5.4	193	2
Chronic Neurological Disease	Yes	191	2.4	217	11.5	408	4.2

Note: * Ischemic heart disease, heart failure, ischemic stroke and high blood pressure.

**Table 2 ijerph-19-08709-t002:** Univariable survival for the outcome of death from COVID-19 (Kaplan Meier and Log-Rank test) in hospitalized patients in Espírito Santo, Brazil, in the period from 1 March 2020 to 31 July 2021.

Dependent Variable (Death from COVID-19)		*p*-Value *	OR	95% CI	
				Lower Limit	Upper Limit
Work-related infection	No	<0.001	3.80	1.89	7.65
	Yes	-	1	-	-
Age group	0–59 years	-	1	-	-
	60–79 years	<0.001	3.59	2.94	4.39
	80 years or older	<0.001	8.30	6.40	10.77
Education	Illiterate	0.018	1.42	1.06	1.89
	Elementary School	0.061	1.34	0.99	1.81
	High School	0.002	1.68	1.21	2.32
	Higher Education	-	1	-	-
Chronic Cardiovascular Disease	No	-	1	-	-
	Yes	<0.001	1.73	1.45	2.07
Chronic Kidney Disease	No	-	1	-	-
	Yes	0.001	2.66	1.68	4.19
Immunodeficiency	No	-	1	-	-
	Yes	<0.001	4.4	1.78	11.55
Smoking	No	-	1	-	-
	Yes	<0.001	2.73	1.90	3.91
Obesity	No	-	1	-	-
	Yes	<0.001	8.33	6.69	10.8
Neoplasms	No	-	1	-	-
	Yes	<0.001	5.40	3.27	8.93
Chronic Neurological Disease	No	-	1	-	-
	Yes	<0.001	5.3	4.00	8.50

***** Significant if *p* < 0.05.

**Table 3 ijerph-19-08709-t003:** Assessment of the risk proportionality assumption by the Cox model.

Variables	Qui-Square	*p*-Value *
Work-related Infection	0.16	0.694
Age Group	7.86	0.020
Education	16.46	0.001
Chronic Cardiovascular Disease **	7.44	0.006
Chronic Kidney Disease	5.81	0.016
Immunodeficiency	0.29	0.590
Smoking	19.4	<0.001
Obesity	22.9	<0.001
Neoplasms	0.94	0.333
Chronic Neurological Disease	16.9	<0.001

***** Significant if *p* < 0.05 based on Pearson’s chi-square test. ****** Ischemic heart disease, heart failure, ischemic stroke, and high blood pressure.

**Table 4 ijerph-19-08709-t004:** Univariable survival for the outcome of death from COVID-19 (Kaplan Meier and Log-Rank test) in hospitalized patients in Espírito Santo, Brazil, in the period from 1 March 2020 to 31 July 2021.

Variables	Categories	Survival Rate (Days)		Mean	±Standard Deviation	Lower Limit	Upper Limit	Log-Rank *p*-Value
		N	%					
Work-related Infection	Yes	183	94.3	121.2	23.2	75.7	166.7	<0.001
	No	5582	78.4	39.5	1.3	37.0	42.0	
	General	5765	78.8	44.9	2.9	39.2	50.6	
Age Group	0–59 years	4494	91.2	71.0	3.3	64.5	77.5	<0.001
	60–79 years	2409	73.7	45.9	3.8	38.4	53.4	
	80 years or older	672	57.0	26.6	1.6	23.4	29.8	
	General	7575	80.8	48.9	3.2	42.7	55.1	
Education	Illiterate	77	72.5	28.8	2.8	23.2	34.3	0.001
	Elementary School	565	72.8	38.9	2.2	34.7	43.1	
	High School	323	82.0	37.7	1.9	33.9	41.5	
	Higher Education	200	78.7	40.8	4.4	32.2	49.4	
	General	1165	77.1	44.5	3.0	38.7	50.4	
Obesity	Yes	483	53.3	35.7	3.6	28.7	42.7	0.001
	No	7068	83.7	47.6	1.7	44.3	50.9	
	General	7551	80.7	48.2	3.0	42.3	54.1	
Neoplasms	Yes	83	47.4	24.5	2.4	19.9	29.2	<0.001
	No	7478	81.4	49.3	3.1	43.1	55.4	
	General	7561	80.8	48.2	3.0	42.4	54.1	
Chronic Neurological Disease	Yes	180	46.3	28.8	2.8	23.2	34.4	<0.001
	No	7380	82.3	52.8	3.4	46.1	59.5	
	General	7560	80.8	48.2	3.0	42.4	54.1	
Chronic Cardiovascular Disease *	Yes	2675	69.9	37.2	1.4	34.4	40.0	<0.001
	No	4883	88.3	61.6	5.7	50.4	72.8	
	General	7558	80.8	48.2	3.0	42.4	54.1	
Chronic Kidney Disease	Yes	135	59.2	34.1	4.5	25.4	42.8	<0.001
	No	7425	81.3	48.5	3.0	42.6	54.4	
	General	7560	80.8	48.2	3.0	42.3	54.1	
Immunodeficiency	Yes	40	59.7	25.7	2.4	21.0	30.4	0.127
	No	7520	80.9	48.5	3.0	42.6	54.4	
	General	7560	80.8	48.2	3.0	42.4	54.1	
Smoking	Yes	171	53.4	35.5	3.6	28.4	42.5	<0.001
	No	7389	81.7	49.1	3.4	42.5	55.8	
	General	7560	80.8	48.2	3.0	42.4	54.1	

Note: * Ischemic heart disease, heart failure, ischemic stroke and high blood pressure.

**Table 5 ijerph-19-08709-t005:** Multivariable survival (crude and adjusted Hazard Ratio by Cox regression model) for the outcome of death from COVID-19 in hospitalized patients in the state of Espírito Santo, Brazil, from 1 March 2020 to 31 July 2021.

Variables		*p*-Value	HR ^1^	95%CI to HR		*p*-Value	HR ^1^	95%CI to HR	
				Lower	Upper			Lower	Upper
Work-related Infection	No	<0.001	5.60	2.90	10.81	<0.001	4.33	2.21	8.46
	Yes	-	1	-	-	-	1	-	-
Age Group **	0–59 years	-	1	-	-	-	1	-	-
	60–79 years	<0.001	2.16	1.89	2.48	<0.001	1.62	1.38	1.89
	80 years or older	<0.001	4.33	3.59	5.23	<0.001	2.56	2.13	3.08
Education **	Illiterate	0.025	1.35	1.04	1.76	0.105	1.23	0.96	1.58
	Elementary School	0.016	1.22	1.04	1.43	0.142	1.22	0.94	1.59
	High School	0.667	0.96	0.80	1.15	0.128	1.24	0.94	1.64
	Higher Education	-	1	-	-	-	1	-	-
Obesity **	No	-	1	-	-	-	1	-	-
	Yes	<0.001	2.72	2.31	3.20	<0.001	2.28	1.99	2.60
Neoplasms *	No	-	1	-	-	-	1	-	-
	Yes	<0.001	1.99	1.62	2.43	<0.001	1.81	1.40	2.35
Chronic Neurological Disease **	No	-	1	-	-	-	1	-	-
	Yes	<0.001	3.05	2.44	3.80	<0.001	1.68	1.39	2.02
Chronic Cardiovascular Disease **	No	-	1	-	-	-	1	-	-
	Yes	<0.001	2.20	1.90	2.50	0.028	118	1.02	1.37
Chronic Kidney Disease **	No	-	1	-	-	-	1	-	-
	Yes	<0.001	1.73	1.41	2.11	0.004	1.50	1.14	1.99
Immunodeficiency *	No	-	1	-	-	-	1	-	-
	Yes	0.134	1.34	0.91	1.96	0.379	1.24	0.78	1.99
Smoking **	No	-	1	-	-	-	1	-	-
	Yes	<0.001	2.52	1.95	3.26	0.001	1.41	1.15	1.75

* Cox model; HR—Hazard Ratio; ^1^ reference category; significant if *p* < 0.05. ** Cox model with time-dependent covariate.

## Data Availability

Not applicable.

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
