# Peer review of "Analysis of Survival of Patients Hospitalized with COVID-19 in Espírito Santo, Brazil"

_ijerph, 2022, doi:10.3390/ijerph19148709_

Round 1
Reviewer 1 Report
The subject is very interesting, the manuscript is well conceived, but some phrases are not very clear and must be reformulated.
I have inserted into the text some of my observations. (marked in blue colour)
English language needs to be seriously revised

Author Response
Comments and Suggestions for Authors
The subject is very interesting, the manuscript is well conceived, but some phrases are not very clear and must be reformulated.
I have inserted into the text some of my observations. (marked in blue colour)
Response: Thank you very much for your careful review. We have really appreciated it!
English language needs to be seriously revised
Response: As suggested, we have done it. We have paid an English proofreading to a bilingual professional, English teacher and Ph.D. in Literature (English Language and English and North American Literature), for polishing the text and correct grammar and spelling mistakes according to the cultured language. An extensive review of English was carried out.

Reviewer 2 Report
The paper reports on the survival of patients hospitalized with COVID-19 in the state of Espírito Santo, Brazil. The factors associated with a lower survival rate were non-work-related infection, age 60-79 and >79 years, fever, and chronic cardiovascular disease. Results are interesting, and they can be considered an epidemiological picture of the state of Espírito Santo during the pandemic (including only the period from March 1, 2020, to July 31, 2021).
Some comments are present:
- Please update the list of references, adding more recent publications about the survival of patients with SARS-CoV2 infection
- English language and style should be improved. Sometimes too long sentences are present
- Please specify that it is a multicenter observational study
-Are there any significant differences between groups in table 1 (Sociodemographic and clinical characterization of hospitalized patients)?
- Please specify which chronic cardiovascular diseases are?
- Please make the graphs more legible. You could add some numbers like survival rates
- How do you explain the results for non-work-related infections?
- Regarding the discussion, I suggest adding some clinical implications of your findings and what your analysis contributes to the various published data on COVID -19.
Author Response
Comments and Suggestions for Authors
The paper reports on the survival of patients hospitalized with COVID-19 in the state of Espírito Santo, Brazil. The factors associated with a lower survival rate were non-work-related infection, age 60-79 and >79 years, fever, and chronic cardiovascular disease. Results are interesting, and they can be considered an epidemiological picture of the state of Espírito Santo during the pandemic (including only the period from March 1, 2020, to July 31, 2021).
Some comments are present:
- Please update the list of references, adding more recent publications about the survival of patients with SARS-CoV2 infection
Response: References have been updated. We have updated references 12 and 25 and those published in year 2022.
Anderegg, N., Panczak, R., Egger, M. et al. Survival among people hospitalized with COVID-19 in Switzerland: a nationwide population-based analysis. BMC Med 20, 164 (2022). https://doi-org.proxy1.library.jhu.edu/10.1186/s12916-022-02364-7
Seif, M., Sharafi, M., Ghaem, H. et al. Factors associated with survival of Iranian patients with COVID-19: comparison of Cox regression and mixture cure model. Trop Dis Travel Med Vaccines 8, 4 (2022).
- English language and style should be improved. Sometimes too long sentences are present
Response: Language and style have been improved. We have paid a bilingual professional, who is an English teacher and Ph.D. in Literature (English Language and English and North American Literature), to proofreading the paper, thus polishing it and correcting its grammar and spelling mistakes, according to the cultured language. An extensive review of English has been carried out.
- Please specify that it is a multicenter observational study
Response: Actually, this is not a multicenter observational study.
-Are there any significant differences between groups in table 1 (Sociodemographic and clinical characterization of hospitalized patients)?
Response: Thanks for this valuable comment. In fact, we had presented only the most representative variables to make the table smaller. Yes, there are. Those with statistical significance have been added to Table 2 (new Table 2). Variables with significant differences are: Work-related infection; Age group; Education; Fever; Difficulty breathing; O2 saturation < 95%; Chronic cardiovascular disease; Chronic kidney disease; Immunodeficiency; and Smoking.
- Please specify which chronic cardiovascular diseases are?
Response: Chronic cardiovascular diseases include: Ischemic heart disease, heart failure, ischemic stroke and high blood pressure. We have added this as a note to the table.
- Please make the graphs more legible. You could add some numbers like survival rates
Response: We have incorporated the suggestions. Graphics have been improved and numbers (survival rates) have been added. Many thanks!
- How do you explain the results for non-work-related infections?
Response: Work-related infections are based on the assumption that the determinants of this disease are related to the physical and social environment of work. Thus, a variety of diseases can be related to work, as long as they find favorable conditions for their occurrence in the environments. However, there is a major limitation in official notification systems as they are insensitive to identifying work-related infections and deaths. Thus, the importance of a complementary epidemiological investigation to identify situations of exposure to SARS-Cov-2 at work is highlighted, which was not possible in the present study. Indeed, we consider this to be a limitation of our study.
Although it is the most common data source for studies with survival analysis, a secondary database is prone to information bias. It is believed that, due to the high turnover during the pandemic, untrained professionals may have failed to understand the variables of the notification form when filling it out and, in this way, promoted inconsistencies in the information, especially those that discuss the work-related infections.
- Regarding the discussion, I suggest adding some clinical implications of your findings and what your analysis contributes to the various published data on COVID -19.
Response: As a strength and implications of this study, our results provide further evidence that there is an increased risk of death for patients with fever, elderly patients with chronic cardiovascular disease, as such factors result in a worsening of the patient's condition. Undoubtedly, the identification of factors associated with the severity of the disease can effectively facilitate the management and provide assistance in order to improve the patients' survival.

Reviewer 3 Report
This study is a retrospective population based observational study. The study is straightforward and clear, well designed and properly analysed. The goals are clearly stated and the data adequate to purse them.
I would like to submit few remarks to the authors.
How the proportionality assumption for the Cox model has been tested, i.e. by graphical inspection or using a formal statistical test. It is generically clear it has been carried out. This could be added in the manuscript.
In Table 3, the “HR” should be reported instead of “RR”. You are calculating how the covariates change the rate of experiencing a failure (death) and not how they change risk. This is coherent with the survival setting of your study.
Work-related infection is always a bit controversial, despite important and interesting, so that it is fundamental to clarify its meaning. It's generally considered debatable if an infection is to be considered work-related. I think this feature should be discussed or emphasized.
Minor details, nonetheless important:
Mean length of stay (line 110) should mention "days".
Editing of Table 3 should be improved. The headings line should avoid words written using two lines.
Author Response
Comments and Suggestions for Authors
This study is a retrospective population based observational study. The study is straightforward and clear, well designed and properly analysed. The goals are clearly stated and the data adequate to purse them.
I would like to submit few remarks to the authors.
How the proportionality assumption for the Cox model has been tested, i.e. by graphical inspection or using a formal statistical test. It is generically clear it has been carried out. This could be added in the manuscript.
Chart 1. Assessment of the proportional risk assessment (premise) by the Cox model.
|
Variables |
Qui-Square |
p-value* |
|
|
Work-related infection |
0.16 |
0.694 |
|
|
Age Group |
7.86 |
0.020 |
|
|
Education |
16.46 |
0.001 |
|
|
Fever |
1.43 |
0.232 |
|
|
Difficulty breathing |
28.32 |
< 0.001 |
|
|
O2 saturation < 95% |
23.52 |
< 0.001 |
|
|
Chronic cardiovascular disease** |
7.44 |
0.006 |
|
|
Chronic kidney disease |
5.81 |
0.016 |
|
|
Immunodeficiency |
0.290 |
0.590 |
|
|
Smoking |
19.40 |
< 0.001 |
*Significant if p<0.05 based on Pearson's chi-square test.
**Ischemic heart disease, heart failure, ischemic stroke and high blood pressure.
In Table 3, the “HR” should be reported instead of “RR”. You are calculating how the covariates change the rate of experiencing a failure (death) and not how they change risk. This is coherent with the survival setting of your study.
Response: As suggested, we have replaced it. Thank you!
Work-related infection is always a bit controversial, despite important and interesting, so that it is fundamental to clarify its meaning. It's generally considered debatable if an infection is to be considered work-related. I think this feature should be discussed or emphasized.
Response: We have added a paragraph addressing this issue in the discussion section before the last paragraph on the limitations of the study. Work-related infections are based on the assumption that the determinants of this disease are related to the physical and social environment of work. Thus, a variety of diseases can be related to work, as long as they find favorable conditions for occurring in the environments. However, there is a major limitation in official notification systems since they are insensitive to identifying work-related infections and deaths. Hence, the importance of conducting a complementary epidemiological investigation to identify situations of exposure to SARS-Cov-2 at work is highlighted, which was not possible in the present study. Indeed, we consider this to be a limitation of our study.
Minor details, nonetheless important:
Mean length of stay (line 110) should mention "days".
Response: Thank you! Your suggestion has been accepted and we have added “days”.
Editing of Table 3 should be improved. The headings line should avoid words written using two lines.
Response: We have done this. Thank you!
All co-authors have agreed to the resubmission with these revisions. The manuscript is consistent with the Guidelines for Authors of the IJERPH.
We are looking forward to hearing from you soon, and we hope to have our manuscript accepted for publication in this renowned journal.
Yours Sincerely,
The authors

Round 2
Reviewer 2 Report
I looked through the revision and found that all the arguments have been addressed correctly. Therefore the paper can be accepted as it is.
Author Response
Response: Thank you very much for your positive feedback.